# Alzheimer’s Disease Connected Genes in the Post-Ischemic Hippocampus and Temporal Cortex

**DOI:** 10.3390/genes13061059

**Published:** 2022-06-14

**Authors:** Ryszard Pluta

**Affiliations:** Laboratory of Ischemic and Neurodegenerative Brain Research, Mossakowski Medical Research Institute, Polish Academy of Sciences, 02-106 Warsaw, Poland; pluta@imdik.pan.pl; Tel.: +48-22-6086-540

**Keywords:** brain ischemia, hippocampus, temporal cortex, genes, Alzheimer’s disease, *amyloid protein precursor*, *β-secretase*, *presenilin 1* and *2*

## Abstract

It is considered that brain ischemia can be causative connected to Alzheimer’s disease. In the CA1 and CA3 regions of the hippocampus and temporal cortex, genes related to Alzheimer’s disease, such as the *amyloid protein precursor (APP)*, *β-secretase* (*BACE1*), *presenilin 1* (*PSEN1*) and *2* (*PSEN2*), are deregulated by ischemia. The pattern of change in the CA1 area of the hippocampus covers all genes tested, and the changes occur at all post-ischemic times. In contrast, the pattern of gene changes in the CA3 subfield is much less intense, does not occur at all post-ischemic times, and is delayed in time post-ischemia relative to the CA1 field. Conversely, the pattern of gene alterations in the temporal cortex appears immediately after ischemia, and does not occur at all post-ischemic times and does not affect all genes. Evidence therefore suggests that various forms of dysregulation of the *APP*, *BACE1* and *PSEN1 and PSEN2* genes are associated with individual neuronal cell responses in the CA1 and CA3 areas of the hippocampus and temporal cortex with reversible cerebral ischemia. Scientific data indicate that an ischemic episode of the brain is a trigger of amyloidogenic processes. From the information provided, it appears that post-ischemic brain injury additionally activates neuronal death in the hippocampus and temporal cortex in an amyloid-dependent manner.

## 1. Introduction

Constantly emerging research of local or global brain ischemia (BI) provides evidence that ischemic damage is likely to be related to the etiology of Alzheimer’s disease (AD) [1,2,3,4,5,6,7,8,9,10,11,12,13,14,15]. Stroke in clinic is a serious life-threatening vascular disease with number of complications such as: cognitive deficits, physical disability and dementia [16,17,18,19,20,21,22]. Both disease have an enormous socio-economic impact all over the world [23,24]. The annual cost of caring and treating for stroke persons in Europe in 2010 was assessed at about EUR 64 billion [25]. In the UK, BI treatment and loss of productivity are estimated at around £ 8.9 billion per year, which is around 5% of the country’s total National Health System budget [26].

### 1.1. BI versus AD

Harvested proof suggests that there is remarkable parallelism between the neuropathogenesis of AD and animal and human BI. First, epidemiological investigations have presented that AD is a factor contributing to the development of BI and vice versa [27,28,29]. Both disease entities, i.e. AD and BI, are characterized by cerebral amyloid angiopathy [30]. Second, brain ischemia and AD have shared risk factors such as: hyperlipidemia, hypertension, obesity and diabetes [31,32]. Third, existing proofs indicate that BI may stimulate the development of AD by triggering the generation and deposition of the amyloid and modifying the tau protein [1,2,4,8,9,10,25,33,34,35]. Fourth, it is believed that inflammation of the brain caused by the immune system plays an important function in the progress and development of AD and BI [15,17,18,19,21,22]. Finally, research shows that tau protein modification is also a key factor in post-ischemia and causes tau protein-dependent neuronal death [9,10,25,34,35,36]. Together, these types of evidence point to common genomic and proteomic risk factors for BI and AD.

### 1.2. Alterations of Hippocampus and Temporal Cortex Post-Ischemia

Post-ischemic brain neurodegeneration in experimental studies leads to molecular and structural alterations in different brain areas, first in the hippocampus and temporal cortex, indicating damages are identical to those in AD [1,2,4,13,37,38,39,40]. BI is the second naturally happening pathology after AD, which triggers predominantly the death of pyramidal neuronal cells in the CA1 area of the hippocampus [2,4]. Post-ischemic hippocampus is considered to be the main neuronal area underlying the impairment of episodic memory, which is the earliest and most visible clinical symptom before dementia following ischemia with AD phenotype [3,30,41,42,43,44,45]. Also, ischemia is responsible for serious damage to the temporal cortex [37], which is the target region of the main axonal output network from the hippocampus. These areas are structurally and functionally connected to each other and are important for learning and memory phenomena [3,30,41,42,43,44,45]. The connotation of increased risk of dementia following ischemic brain injury with age, atrophy of the hippocampus and neurodegenerative damages in the temporal cortex with hemorrhages has been observed [2,30,37,46,47,48].

### 1.3. Amyloid in Post-Ischemic Brain

Rats following BI presented intracellular staining to the β-amyloid peptide, to the N-terminal of amyloid protein precursor (APP) and at the C-terminal of APP in the brain [1,4]. The accumulation of diverse fragments of the amyloid protein precursor in extracellular space was mainly noted following brain ischemia in human and animal hippocampus, in the form of irregularly dispersed diffuse and senile amyloid plaques [1,2,4,49,50,51,52,53,54]. Current facts about the activation of genes and proteins related with AD after ischemia, and the neuropathology of both AD and BI, point to the situation that analogous processes contribute to the death of neurons and the disintegration of the brain parenchyma in both diseases, finally leading to development of dementia [34,35,55,56,57,58,59,60,61]. The incidence of BI indicates that the vascular system is a probable factor creating degeneration and dementia in AD. 

This review presents the latest knowledge on the function of genes involved in the amyloidogenic processing of the APP, which is related to the production and deposition of amyloid in the hippocampus and temporal cortex following BI. It was also considered whether the signaling pathway of the APP is involved in inducing neuronal death in the hippocampus and temporal cortex in an amyloid dependent manner.

## 2. mRNAs Related with the Post-Ischemic APP

Due to the limited amount of new information in animal studies on damage to the APP after BI, this section of the review presents the first stages in mRNA research linked to the processing of the APP following different models of BI. This indicates that there is a serious need for evidence for a new causal neuropathological role for amyloid in BI; which substance is most likely to ultimately have an irreversible consequence on ischemic outcome.

### 2.1. mRNA of the APP

After transient experimental local ischemic brain injury, the mRNA level of the APP was raised both in the penumbra and in the core, by 200 and 150%, respectively, over 7 days post-ischemia [62,63]. Furthermore, following permanent focal BI injury, the mRNA domain of the Kunitz-type protease inhibitor domain-containing APP in the brain cortex was increased for 21 days [64]. Additionally, following temporary focal BI, the APPs, 751 and 770 mRNAs, were raised during 7 days of reperfusion [65]. Only the APP-695 is existing in the neuronal cells, therefore it should be presumed that it has been degraded or absent due to the death of neuronal cells in the hippocampus and temporal cortex following cerebral ischemia. Moreover, 1 h following focal BI injury in ovariectomized animals, the raised mRNA level of the APP was noted in all ischemic brain parts [62]. But, estrogen therapy decreases the mRNA level of the APP in post-ischemic brain [62]. These results suggest that estrogen treatment can be used to decrease the mRNA of the APP following the ischemic incident.

### 2.2. mRNA of Enzymes Metabolizing the APP

The APP is processed by α-secretase, and this phenomenon is a non-amyloidogenic pathway. After experimental focal and global ischemic brain injury, the level of α-secretase mRNA and gene expression decreases, including in the hippocampus [61,66,67]. The second phenomenon is called the amyloidogenic route, and the APP is cleaved by β- and γ-secretase as a result of this reaction, β-amyloid peptide is formed (Figure 1) [6]. Some investigations have documented that ischemic incidence of the brain activates β-secretase post-ischemia [68,69,70,71]. Alternative investigation showed alterations in mRNA levels of three enzymes that metabolize the APP: β-secretase, glutaminyl cyclase and cathepsin B, which were raised in the hippocampus and cortex post-ischemia [72].

Three days post-ischemia, the highest level of presenilin 1 (PSEN1) mRNA was noted in the neurons of CA3 region of the hippocampus [73]. This evidence suggests that a raised level of PSEN1 mRNA probably is connected with the answer of neurons to BI. In an additional study, the raised level of PSEN1 mRNA presented the maximum increase in the cortex, striatum, and hippocampus following local BI injury [74]. In the above research, the raised level of PSEN1 mRNA was greater on the opposite side to focal BI changes. This phenomenon may imitate the death of brain neuronal cells on the ipsilateral side. The mRNA of PSEN1, which was raised following BI [73,74], is involved in the generation of the β-amyloid peptide by the γ-secretase complex (Figure 1) [6,75]. The above evidence helps to understand the gradual neuronal loss after the ischemic injury of the brain and the silent, delayed deposition of the β-amyloid peptide in the post-ischemic brain (Figure 1) [4,11].

## 3. Genes Engaged in the Generation of Amyloid in the Post-Ischemic Hippocampus and Temporal Cortex

In rodents surviving 2 days to 2 years after BI, intra- and extracellular deposition of various fragments of the APP in the hippocampus and temporal cortex were noted (Figure 1) [1,2,33,76,77,78,79,80,81]. Deposition of different fragments of the APP was always observed in neurons and glia [2,4,32,79,80,82,83,84,85]. Data indicate that astrocytes, which accumulate huge amounts of the amyloid, are implicated in the generation of glial scar [2,32,84]. In addition, astrocytes with disproportionate amyloid increase may be involved in restoring the hippocampus post-ischemia, which finally leads to the death of astrocytes [2,4,32,86]. The extracellular deposition of amyloid showed features of diffuse and senile amyloid plaques [2,4,52,80]. Deposition of amyloid in neurons and astrocytes is a symptom of neuropathological processing of the APP in the course of ischemic neurodegeneration of the hippocampus and other brain structures [4,78,84,87,88]. The evidence clearly confirmed that the deposition of amyloid post-ischemia in the hippocampus and other brain parts is responsible for the secondary neurodegenerative mechanisms that cause gradual death of ischemic neuronal cells, which additionally influences the post-ischemic outcome (Figure 1) [4,42,79,80,85,89,90]. Senile and diffuse amyloid plaques have also been documented in the hippocampus in patients with a history of BI [49,50,51,53,91]. Increased accumulation of different amyloids contributes to the advancement of post-ischemic neurodegenerative pathways and in the end to the development of AD dementia (Figure 1). Furthermore, clinical studies have revealed a rise in the level of amyloid in the serum in patients with a history of BI [92,93,94]. Increased blood amyloid in these patients was found to be harmfully correlated with neurological outcomes post-ischemia [93]. The above evidence indicate that after ischemia, the generated amyloid is additionally responsible for the progression of neurodegeneration that worsen the outcome post-ischemia through neuronal death (Figure 1) [14].

### 3.1. CA1 Area of Hippocampus

The expression of the *APP* gene in the CA1 area of the hippocampus was decreased 2 days post-ischemia and increased above the control values during 7–30 days (Table 1) [55]. The *β-secretase* (*BACE1*) and *presenilin 1* and *2 (PSEN2*) genes were upregulated between 2 and 7 days and were decreased at 30 days post-ischemia (Table 1) [55].

### 3.2. CA3 Area of Hippocampus

In the CA3 region, at 2 and 30 days after ischemia, *APP* gene expression was around control values (Table 1) [61], but, 7 days after ischemia, *APP* gene expression was beyond control values (Table 1) [61]. Expression of the *BACE1* gene in the CA3 area was below the control values at 2 and 7 days, while 30 days following ischemia, it was above the control values (Table 1) [61]. Expression of the *PSEN1* gene was above control values at 2 and 7 days, and fluctuated around the control values 30 days post-ischemia (Table 1) [61]. Following ischemia *PSEN2* expression fluctuated around the control values at 2 days, decreased on day 7 and was above the control values on day 30 (Table 1) [61].

### 3.3. Temporal Cortex

The expression of the *APP* gene in the cortex was reduced 2 days post-ischemia and increased above the control values between 7 and 30 days (Table 1) [56]. Expression of the *BACE1* gene in the above area was upregulated at 2 days, while at 7 and 30 days following ischemia, it oscillated around the control values (Table 1) [56]. The *PSEN1* gene fluctuated around the control values 2, 7 and 30 days after ischemia (Table 1) [57]. Expression of the *PSEN2* gene was the above control values at 2 days, and oscillated around the control values at 7 and 30 days after ischemia (Table 1) [57].

## 4. Conclusions

The pattern of dysregulation of genes linked with AD in the CA3 region of the hippocampus post-ischemia is much slower in time and less intense than that occurring in the CA1 area of the hippocampus. Thus, gene changes in the CA1 area of the hippocampus indicate a faster progression of post-ischemic pathology compared to the CA3 area. On the other hand, the course of events in the cortex is the slowest and less marked compared to those in the CA1 and CA3 regions in the early period after hippocampal ischemia with the complete absence of changes in the expression of *BACE1, PSEN1* and *PSEN2* genes in the later periods of recirculation. These data clearly indicate that the occurrence of cerebral ischemia triggers amyloidogenic processes that are extremely dangerous for the survival of the brain (Figure 1).

The relationship of ischemic damages in neuronal cells with disturbed β-amyloid peptide homeostasis has similarly been documented by immunohistochemistry following temporary focal or global BI in animals and humans [1,2,4,49,50,51,53,54]. Based on the research and analyses presented here, it appears that cerebral ischemia is an event that triggers the amyloidogenic processing of the APP, the products of which, in particular amyloid, are involved in the amyloidogenic phenomenon and irreversible damage to neurons post-ischemia (Figure 1). The relationship between ischemic neuronal death and the expansion of the amyloidogenic metabolism of APP after BI is rather certain (Figure 1 and Figure 2). Recent evidence undoubtedly points to a possible neuropathogenic interaction between post-ischemic neurons and the ischemic amyloidogenic metabolism of the APP to amyloid, a process that is characteristic of the development of the AD (Figure 2). Raised expression of *presenilin* genes especially in CA1 region and, consequently, elevated production of their proteins may stimulate neurodegeneration following BI injury by rising neuronal cells sensitivity to ischemia [95]. Presenilins disturb the homeostasis of calcium in neurons, which leads to increased susceptibility of them to apoptosis [4]. A raised level of the soluble amyloid in the brain and in the blood [92,93,94] post-ischemia inclines neuronal cells to apoptosis, too. Additionally, amyloid stimulates post-ischemia hyperphosphorylation of tau protein, leading to a vicious cycle (Figure 1 and Figure 2) [96]. Definitively, the amyloid influences the phosphorylation of the tau protein post-ischemia, which increases apoptosis leading to a chain reaction [35,96]. Furthermore, it is accepted as true that the rise in the generation of amyloid in the early phase post-ischemia influences the sealing of the blood-brain barrier and healing of sites after vanished away neuronal cells [97,98,99,100]. Chronic generation of amyloid is likely to undergo a pathological healing processes followed by durable permeability to the blood-brain barrier and deposition of amyloid in the hippocampus, and then spreading into the cortex and lastly into the entire brain (Figure 2) [48]. Studies show that ischemia leading to increased amyloid accumulation in the brain strongly supports the notion that the damage to ischemic neuronal cells is primary, while the observed increase in amyloid accumulation is rather a secondary phenomenon due to changes in the membranes of neurons and vessels [55,77].

Expression of the *APP* gene does not overlap with previous results on the staining of different parts of the APP in the CA1 area and temporal cortex two days post-ischemia [1,4]. Results indicate that there is a disagreement between the expression of the *BACE1* gene and the *APP*, whose expression has been decreased below the control values in the ischemic CA1 region and temporal cortex two days post-ischemia. It is clear that necrotic neuronal death related with acute post-ischemic neuropathology prevails at this time [2,4,101,102]. During necrotic death of neuronal cells, the discontinuity of cell membranes, the membranes of which are rich in amyloid protein precursor, has been shown [103,104]. The APP is abundant in cell membranes [105], so in the above state there is an overload with the protease substrate, which is the APP. The discontinuity of cell membranes, in particular by neuronal cells, permits necrotic neurons to cause uncontrolled release and processing of the APP [106].

In the CA1 area, the presented information shows that in the following days the expression of the *BACE1* gene increases. There were exactly the opposite changes in the CA3 region. In contrast, alterations in the temporal cortex oscillated around control values. At the same time, an increase was observed in the expression of the *APP* gene in the CA1 field and temporal cortex following BI injury. In the region of CA3, an increase in the expression of APP was noted only 7 days after BI. However, the observations are consistent with the strong staining of different fragments of the APP after BI [1,2,4]. Over 30 days post-ischemia, widespread neuronal cells loss usually ends in the CA1 subfield and in the layers 3, 5 and 6 of the temporal cortex, and during this time the expression of the *APP* gene and its product increases (Figure 2) [55,56,77]. On the other hand, neurons death in the CA3 area occurs several months later after BI [4].

The β- and γ-secretase action leads to the production of amyloid peptides, which may cause secondary and final injury to ischemic neuronal cells in brain (Figure 1). The presented observations show that ischemia of the hippocampus and cortex does not touch the expression of secretases for all times post-ischemia and does not rise the amyloidogenesis in the hippocampus and temporal cortex all the time post-ischemia. The obtained results indicate a new, complicated role of the examined genes, which are related with AD, in the post-ischemic hippocampus and cortex. It can be assumed that we can distinguish between focal and global changes in the processing of the APP. This phenomenon is probably connected with the transfer of soluble amyloid peptides from the plasma to the brain parenchyma following ischemic brain episode [92,93,94,107]. 

It is known that age-connected vascular alterations go together with or even go before the development of AD, which makes it highly probable that they may play a key pathogenic role. While the pathways of these changes remains to be determined, amyloid is an important pathogenic factor, but not unique to the brain, as is it outside the brain. Together, the data suggests that vascular pathologies are a highly likely neuropathogenic factor in age-linked dementia, including AD, inextricably linked to disorder beginning and advancement [108]. Therefore, the involvement of vascular aspects in prophylactic, diagnostic and curative methods must be taken into account in order to meet one of the main health tests of our time [108]. 

Recent data define additional and novel mechanisms of pyramidal neurons death in the CA1 and CA3 regions and layers 3, 5 and 6 of the cortex following transient brain ischemia. In vivo monitoring of gene dysregulation using reversible experimental of BI in animals opens the method to a well understanding of the involvement of AD-linked genes and their products to the pathology of AD and the progress of neurodegeneration post-ischemia with dementia (Figure 2). Above evidences will help to understand progressive injury post-ischemia, chronic accumulation of amyloid, and delayed expansion of AD degeneration that extents from hippocampus to the temporal lobe and other parts of the brain tissue [6,55,56,57,61,109,110,111]. What is even more, the observations presented that BI injury starts delayed neuronal loss in the hippocampus and temporal cortex in an amyloid-dependent manner (Figure 1). Thus describing a new and significant mechanism for controlling the survival or death of post-ischemic neurons. In addition, dysfunction of the genes and proteins connected with AD following ischemia ultimately leads to chronic neuropathology with the expansion of AD type dementia (Figure 2). That this is currently an important problem is evidenced by the report summarizing the debate and instructions of the working group established by the National Heart, Lung and Blood Institute and the National Institute of Neurological Disorders and Stroke to assess the condition of the field in the vascular influences to dementia studies and to determine research priorities. As shown in this report, advances in understanding of the molecular processes of vascular contributions to dementia could lead to the elaboration of potential prevention and new cure tactics to decrease the problem of dementia [112]. A well understanding of the social factors of health that affect the risk of both vascular disorder and vascular influence to dementia can provide insight into methods to decrease the gap between developed and developing countries in vascular input to cognitive deficiency and dementia [112].

## Figures and Tables

**Figure 1 genes-13-01059-f001:**
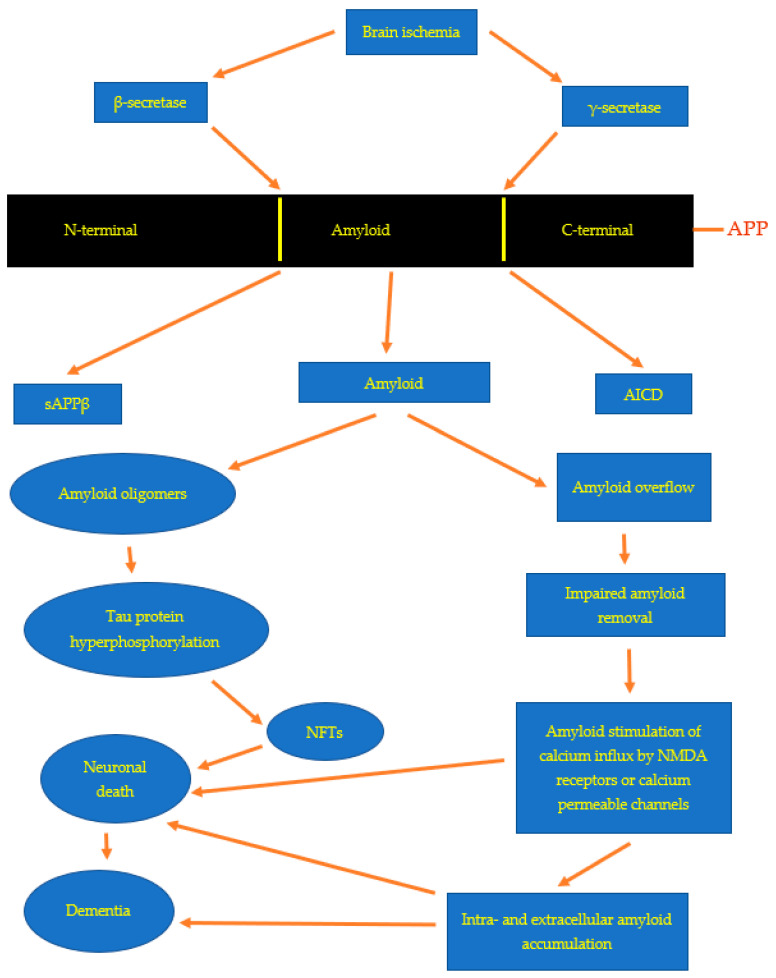
Proposed pathways for amyloid accumulation and toxicity. APP-amyloid protein precursor, sAPPβ-soluble amyloid protein precursor, NFTs-neurofibrillary tangles, AICD-amyloid protein precursor intracellular domain.

**Figure 2 genes-13-01059-f002:**
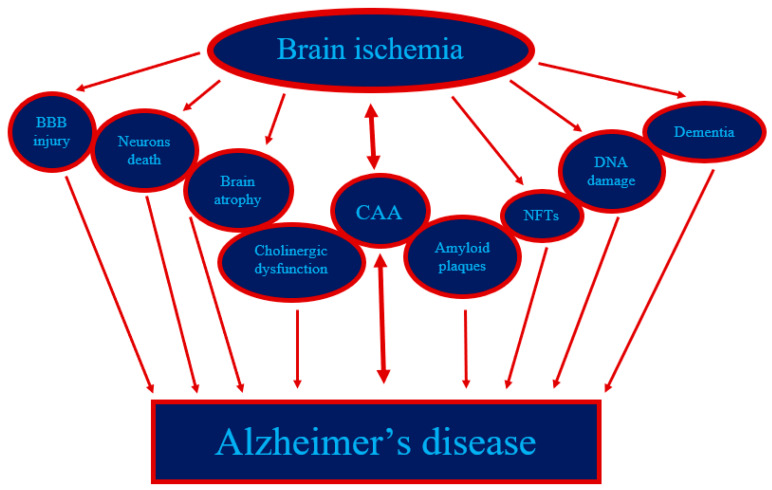
The events in post-ischemic brain injury have a remarkable parallel with Alzheimer’s disease. BBB—blood–brain barrier; CAA—cerebral amyloid angiopathy; NFTs—neurofibrillary tangles.

**Table 1 genes-13-01059-t001:** AD-linked genes in the hippocampus and temporal cortex post-ischemia.

	Genes	*APP*	*BACE1*	*PSEN1*	*PSEN2*		
Survival			Citation
**CA1 area of hippocampus**
**2 days**	↓	↑↑	↑	↑↑	[55]
**7 days**	↑	↑	↑	↑	[55]
**30 days**	↑	↓	↓	↓	[55]
**CA3 area of hippocampus**
**2 days**	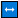	↓	↑	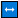	[61]
**7 days**	↑	↓	↑	↓	[61]
**30 days**	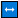	↑	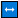	↑	[61]
**Temporal cortex**
**2 days**	↓	↑↑	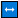	↑↑	[56,57]
**7 days**	↑	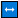	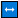	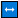	[56,57]
**30 days**	↑	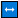	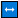	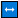	[56,57]

Expression: ↑↑ increase; ↑ increase; ↓ decrease; 
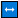
 oscillation around control values. Genes: *APP*-*amyloid protein precursor*, *BACE1*-*β-secretase*, *PSEN1*-*presnilin 1*, *PSEN2*-*presenilin 2*.

## Data Availability

At the author’s for correspondence.

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
