# Peer review of "Alzheimer’s Disease Connected Genes in the Post-Ischemic Hippocampus and Temporal Cortex"

_genes, 2022, doi:10.3390/genes13061059_

Round 1

Reviewer 1 Report

In this article Dr. Pluta describes the role of cerebral ischemia on Alzheimer’s Disease associated genes.

Mayor concerns:

1. This article looks like a research paper with standard format such as Introduction, Results, and Discussion. Generally, the review should have several headings for topics or subcategories. In addition, avoid to use “Discussion” at the end of the article. Instead, a short conclusion section should be added where a future direction is summarized.

2. The abstract is too long and too descriptive for the details of Dr. Pluta’s work. Therefore, the abstract should be rewritten and presented very simply in brief without going into great details.

3. The introduction is too long. It should contain only approximately 10% of the overall review article. Again, the long introduction should be broken down in several topic headings and should refer to the actual concepts covered in that section.

4. The reference list contains excessive self-citations of Dr. Pluta’s works (more than 30% of total references). In addition, Dr. Pluta neglects to citate important works by the other groups, for example vascular contribution of Alzheimer's Disease by Drs. Zlokovic, Iadecola etc.

5. Lines 113, 114: There are 3 isoforms of amyloid precursor proteins: APP-695, APP-751, and APP-770. It is well known that APP-695 is expressed exclusively in neuronal cells. Hence, the author needs to mention in the article what happened with neuronal isoform of APP-695 in the hippocampus and temporal cortex during cerebral ischemia.

6. A schematic presentation of APP expression and processing is missing.

Author Response

Reviewer 1. All changes are in red. In this article Dr. Pluta describes the role of cerebral ischemia on Alzheimer’s Disease associated genes.

Major concerns: 1. This article looks like a research paper with standard format such as Introduction, Results, and Discussion. Generally, the review should have several headings for topics or subcategories. In addition, avoid to use “Discussion” at the end of the article. Instead, a short conclusion section should be added where a future direction is summarized. Done.

2. The abstract is too long and too descriptive for the details of Dr. Pluta’s work. Therefore, the abstract should be rewritten and presented very simply in brief without going into great details. Done.

3. The introduction is too long. It should contain only approximately 10% of the overall review article. Again, the long introduction should be broken down in several topic headings and should refer to the actual concepts covered in that section. Done.

4. The reference list contains excessive self-citations of Dr. Pluta’s works (more than 30% of total references). In addition, Dr. Pluta neglects to citate important works by the other groups, for example vascular contribution of Alzheimer's Disease by Drs. Zlokovic, Iadecola etc. Done. 108 and 112.

5. Lines 113, 114: There are 3 isoforms of amyloid precursor proteins: APP-695, APP-751, and APP-770. It is well known that APP-695 is expressed exclusively in neuronal cells. Hence, the author needs to mention in the article what happened with neuronal isoform of APP-695 in the hippocampus and temporal cortex during cerebral ischemia. Done.

6. A schematic presentation of APP expression and processing is missing. Done.

Reviewer 2 Report

The review article entitled “Alzheimer’s Disease-Associated Genes in the Post-ischemic Hippocampus and Temporal Cortex” is a very interesting work that describes the relationship between the expression of the Alzheimer’s diseases associated genes (amyloid protein precursor, preselinin, β-secretase, etc.) and stroke. This is a novel topic that could provide important information about the mechanisms that regulate the death of post-ischemic neuronal cells.

I consider that it is a well-written manuscript in Standard English. The author properly describes the evidence showing the relation between Alzheimer's and stroke. Nevertheless, I have one comment. In the abstract, the author recapitulated the current results about the topic, I suggest changing it and instead include a concise description of the changes in the expression of the genes and add a description of the mechanism that could be associated with those alterations and the relation with stroke in animals and human. Similarly, in the discussion, the author again mentions those changes (up and downs of the genes) that may not necessary to repeat.

Minor comment: In lines 110 and 111, the author mentions “following permanent focal brain ischemia injury without reperfusion”… When the ischemia is permanent, it does not have reperfusion; therefore, the sentence is redundant.  

Author Response

Reviewer 2. All changes are in red. The review article entitled “Alzheimer’s Disease-Associated Genes in the Post-ischemic Hippocampus and Temporal Cortex” is a very interesting work that describes the relationship between the expression of the Alzheimer’s diseases associated genes (amyloid protein precursor, preselinin, β-secretase, etc.) and stroke. This is a novel topic that could provide important information about the mechanisms that regulate the death of post-ischemic neuronal cells. Thanks. I consider that it is a well-written manuscript in Standard English. The author properly describes the evidence showing the relation between Alzheimer's and stroke. Nevertheless, I have one comment. In the abstract, the author recapitulated the current results about the topic, I suggest changing it and instead include a concise description of the changes in the expression of the genes and add a description of the mechanism that could be associated with those alterations and the relation with stroke in animals and human. Similarly, in the discussion, the author again mentions those changes (up and downs of the genes) that may not necessary to repeat. Done. Minor comment: In lines 110 and 111, the author mentions “following permanent focal brain ischemia injury without reperfusion”… When the ischemia is permanent, it does not have reperfusion; therefore, the sentence is redundant Done.

Round 2

Reviewer 1 Report

None